# OPEN-WORLD SEMI-SUPERVISED LEARNING

**Kaidi Cao**\*, **Maria Brbić**\*, **Jure Leskovec**
Department of Computer Science
Stanford University
{kaidicao, mbrbic, jure}@cs.stanford.edu

## ABSTRACT

A fundamental limitation of applying semi-supervised learning in real-world settings is the assumption that unlabeled test data contains only classes previously encountered in the labeled training data. However, this assumption rarely holds for data in-the-wild, where instances belonging to novel classes may appear at testing time. Here, we introduce a novel *open-world semi-supervised learning* setting that formalizes the notion that novel classes may appear in the unlabeled test data. In this novel setting, the goal is to solve the class distribution mismatch between labeled and unlabeled data, where at the test time every input instance either needs to be classified into one of the existing classes or a new unseen class needs to be initialized. To tackle this challenging problem, we propose ORCA, an end-to-end deep learning approach that introduces uncertainty adaptive margin mechanism to circumvent the bias towards seen classes caused by learning discriminative features for seen classes faster than for the novel classes. In this way, ORCA reduces the gap between intra-class variance of seen with respect to novel classes. Experiments on image classification datasets and a single-cell annotation dataset demonstrate that ORCA consistently outperforms alternative baselines, achieving 25% improvement on seen and 96% improvement on novel classes of the ImageNet dataset.

## 1 INTRODUCTION

With the advent of deep learning, remarkable breakthroughs have been achieved and current machine learning systems excel on tasks with large quantities of labeled data (LeCun et al., 2015; Silver et al., 2016; Esteva et al., 2017). Despite the strengths, the vast majority of models are designed for the closed-world setting rooted in the assumption that training and test data come from the same set of predefined classes (Bendale & Boult, 2015; Boult et al., 2019). This assumption, however, rarely holds for data in-the-wild, as labeling data depends on having the complete knowledge of a given domain. For example, biologists may prelabel known cell types (seen classes), and then want to apply the model to a new tissue to identify known cell types but also to *discover novel* previously unknown cell types (unseen classes). Similarly, in social networks one may want to classify users into predefined interest groups while also discovering new unknown/unlabeled interests of users. Thus, in contrast to the commonly assumed closed world, many real-world problems are inherently open-world — new classes can emerge in the test data that have never been seen (and labeled) during training.

Here we introduce *open-world semi-supervised learning* (open-world SSL) setting that generalizes semi-supervised learning and novel class discovery. Under open-world SSL, we are given a labeled training dataset as well as an unlabeled dataset. The labeled dataset contains instances that belong to a set of *seen classes*, while instances in the unlabeled/test dataset belong to both the seen classes as well as to an unknown number of *unseen classes* (Figure 1). Under this setting, the model needs to either classify instances into one of the previously seen classes, or discover new classes and assign instances to them. In other words, open-world SSL is a transductive learning setting under class distribution mismatch in which unlabeled test set may contain classes that have never been labeled during training, *i.e.*, are not part of the labeled training set.

Open-world SSL is fundamentally different but closely related to two recent lines of work: robust semi-supervised learning (SSL) and novel class discovery. Robust SSL (Oliver et al., 2018; Guo

---

\*The two first authors made equal contributions.

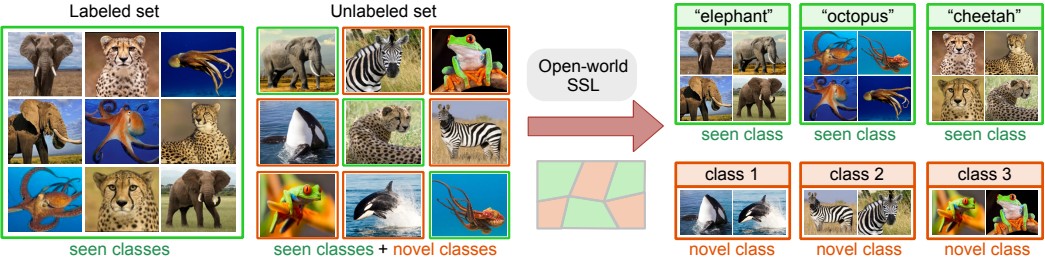

Figure 1: In the open-world SSL, the unlabeled dataset may contain classes that have never been encountered in the labeled set. Given unlabeled test set, the model needs to either assign instances to one of the classes previously seen in the labeled set, or form a novel class and assign instances to it.

et al., 2020; Chen et al., 2020b; Guo et al., 2020; Yu et al., 2020) assumes class distribution mismatch between labeled and unlabeled data, but in this setting the model only needs to be able to recognize (reject) instances belonging to novel classes in the unlabeled data as out-of-distribution instances. In contrast, instead of rejecting instances belonging to novel classes, open-world SSL aims at discovering individual novel classes and then assigning instances to them. Novel class discovery (Hsu et al., 2018; 2019; Han et al., 2019; 2020; Zhong et al., 2021) is a clustering problem where one assumes unlabeled data is composed only of novel classes. In contrast, open-world SSL is more general as instances in the unlabeled data can come from seen as well as from novel classes. To apply robust SSL and novel class discovery methods to open-world SSL, one could in principle adopt a multi-step approach by first using robust SSL to reject instances from novel classes and then applying a novel class discovery method on rejected instances to discover novel classes. An alternative would be that one could treat all classes as "novel", apply novel class discovery methods and then match some of the classes back to the seen classes in the labeled dataset. However, our experiments show that such ad hoc approaches do not perform well in practice. Therefore, it is necessary to design a method that can solve this practical problem in an end-to-end framework.

In this paper we propose ORCA (Open-woRld with unCertainty based Adaptive margin) that operates under the novel open-world SSL setting. ORCA effectively assigns examples from the unlabeled data to either previously seen classes, or forms novel classes by grouping similar instances. ORCA is an end-to-end deep learning framework, where the key to our approach is a novel uncertainty adaptive margin mechanism that gradually decreases plasticity and increases discriminability of the model during training. This mechanism effectively reduces an undesired gap between intra-class variance of seen with respect to the novel classes caused by learning seen classes faster than the novel, which we show is a critical difficulty in this setting. We then develop a special model training procedure that learns to classify data points into a set of previously seen classes while also learning to use an additional classification head for each newly discovered class. Classification heads for seen classes are used to assign the unlabeled examples to classes from the labeled set, while activating additional classification heads allows ORCA to form a novel class. ORCA does not need to know the number of novel classes ahead of time and can automatically discover them at the deployment time.

We evaluate ORCA on three benchmark image classification datasets adapted for open-world SSL setting and a single-cell annotation dataset from biology domain. Since no existing methods can operate under the open-world SSL setting we extend existing state-of-the-art SSL, open-set recognition and novel class discovery methods to the open-world SSL and compare them to ORCA. Experimental results demonstrate that ORCA effectively addresses the challenges of open-world SSL setting and consistently outperforms all baselines by a large margin. Specifically, ORCA achieves 25% and 96% improvements on seen and novel classes of the ImageNet dataset. Moreover, we show that ORCA is robust to unknown number of novel classes, different distributions of seen and novel classes, unbalanced data distributions, pretraining strategies and a small number of labeled examples.

## 2 RELATED WORK

We summarize similarities and differences between open-world SSL and related settings in Table 1. Additional related work is given in Appendix A.

Table 1: Relationship between our novel open-world SSL and other machine learning settings.

| Setting | Seen classes | Novel classes | Prior knowledge |
|---|---|---|---|
| Novel class discovery | Not present | Discover | None |
| SSL | Classify | Not present | None |
| Robust SSL | Classify | Reject | None |
| Generalized zero-shot learning | Classify | Discover | Class attributes |
| Open-set recognition | Classify | Reject | None |
| Open-world recognition | Classify | Discover | Human-in-the-loop |
| **Open-world SSL** | Classify | Discover | None |

**Novel class discovery.** In novel class discovery (Hsu et al., 2018; Han et al., 2020; Brbic et al., 2020; Zhong et al., 2021), the task is to cluster unlabeled dataset consisting of similar, but completely disjoint, classes than those present in the labeled dataset which is utilized to learn better representation for clustering. These methods assume that at the test time all the classes are novel. While these methods are able to discover novel classes, they do not recognize the seen/known classes. On the contrary, our open-world SSL is more general because unlabeled test set consists of novel classes but also classes previously seen in the labeled data that need to be identified. In principle, one could extend novel class discovery methods by treating all classes as "novel" at test time and then matching some of them to the known classes from the labeled dataset. We adopt such approaches as our baselines, but our experiments show that they do not perform well in practice.

**Semi-supervised learning (SSL).** SSL methods (Chapelle et al., 2009; Kingma et al., 2014; Laine & Aila, 2017; Zhai et al., 2019; Lee, 2013; Xie et al., 2020; Berthelot et al., 2019; 2020; Sohn et al., 2020) assume closed-world setting in which labeled and unlabeled data come from the same set of classes. Robust SSL methods (Oliver et al., 2018; Chen et al., 2020b; Guo et al., 2020; Yu et al., 2020) relax the SSL assumption by assuming that instances from novel classes may appear in the unlabeled test set. The goal in robust SSL is to reject instances from novel classes which are treated as out-of-distribution instances. Instead of rejecting instances from novel classes, in open-world SSL the goal is to discover individual novel classes and then assign instances to them. To extend robust SSL to open-world SSL, one could apply clustering/novel class discovery methods to discarded instances. Early work (Miller & Browning, 2003) considered solving the problem in such a way using extension of the EM algorithm. However, our experiments show that by discarding the instances the embedding learned by these methods does not allow for accurate discovery of novel classes.

**Open-set and open-world recognition.** Open-set recognition (Scheirer et al., 2012; Geng et al., 2020; Bendale & Boult, 2016; Ge et al., 2017; Sun et al., 2020a) considers the inductive setting in which novel classes can appear during testing, and the model needs to reject instances from novel classes. To extend these methods to open-world setting, we include a baseline that discovers classes on rejected instances. However, results show that such approaches can not effectively address the challenges of open-world SSL. Similarly, open-world recognition approaches (Bendale & Boult, 2015; Rudd et al., 2017; Boult et al., 2019) require the system to incrementally learn and extend the set of known classes with novel classes. These methods incrementally label novel classes by human-in-the-loop. In contrast, open-world SSL leverages unlabeled data in the learning stage and does not require human-in-the-loop.

**Generalized zero-shot learning (GZSL).** Like open-world SSL, GZSL (Xian et al., 2017; Liu et al., 2018; Chao et al., 2016) assumes that classes seen in the labeled set and novel classes are present at the test time. However, GZSL imposes additional assumption about availability of prior knowledge given as auxiliary attributes that uniquely describe each individual class including the novel classes. This restrictive assumption severely limits the application of GZSL methods in practice. In contrast, open-world SSL is more general as it does not assume any prior information about classes.

## 3 PROPOSED APPROACH

In this section, we first define the open-world SSL setting. We follow by an overview of ORCA framework and then introduce each of the components of our framework in details.

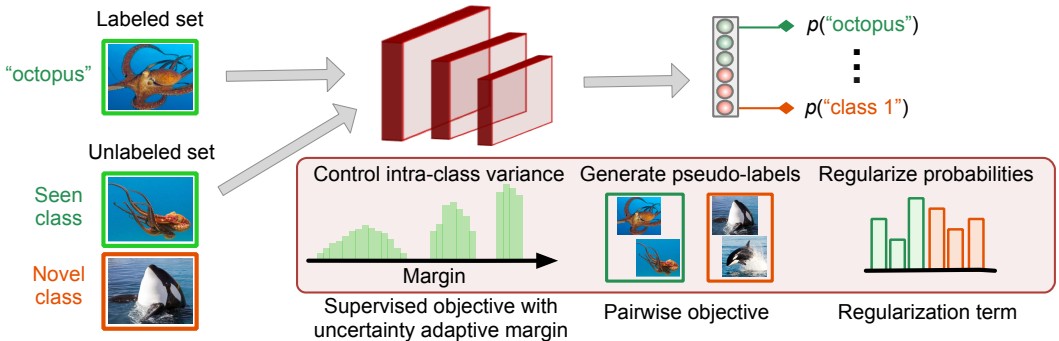

Figure 2: Overview of ORCA framework. ORCA utilizes additional classification heads for novel classes. Objective function in ORCA consists of *(i)* supervised objective with uncertainty adaptive margin, *(ii)* pairwise objective that generates pseudo-labels, and *(iii)* regularization term.

### 3.1 OPEN-WORLD SEMI-SUPERVISED LEARNING SETTING

In the open-world SSL, we assume transductive learning setting in which labeled part of the dataset $\mathcal{D}_l = \{(x_i, y_i)\}_{i=1}^n$ and unlabeled part of the dataset $\mathcal{D}_u = \{(x_i)\}_{i=1}^m$ are given at the input. We denote the set of classes seen in the labeled data as $\mathcal{C}_l$ and the set of classes in the unlabeled test data as $\mathcal{C}_u$. We assume category/class shift, *i.e.*, $\mathcal{C}_l \cap \mathcal{C}_u \neq \emptyset$ and $\mathcal{C}_l \neq \mathcal{C}_u$. We consider $\mathcal{C}_s = \mathcal{C}_l \cap \mathcal{C}_u$ as a set of seen classes, and $\mathcal{C}_n = \mathcal{C}_u \backslash \mathcal{C}_l$ as a set of novel classes.

**Definition 1** (Open-world SSL). In the open-world SSL, the model needs to assign instances from $\mathcal{D}_u$ either to previously seen classes $\mathcal{C}_s$, or form a novel class $c \in \mathcal{C}_n$ and assign instances to it.

Note that open-world SSL generalizes novel class discovery and traditional (closed-world) SSL. Novel class discovery assumes that classes in labeled and unlabeled data are disjoint, *i.e.*, $\mathcal{C}_l \cap \mathcal{C}_u = \emptyset$, while (closed-world) SSL assumes the same classes in labeled and unlabeled data, *i.e.*, $\mathcal{C}_l = \mathcal{C}_u$.

### 3.2 OVERVIEW OF ORCA

The key challenge to solve open-world SSL is to learn both from seen/labeled as well as unseen/unlabeled classes. This is challenging as models learn discriminative representations faster on the seen classes compared to the novel classes. This leads to smaller intra-class variance of seen classes compared to novel classes. To circumvent this problem, we propose ORCA, an approach that reduces the gap between intra-class variance of seen and novel classes during training using the uncertainty adaptive margin. The key insight in ORCA is to control intra-class variance of seen classes using uncertainty on unlabeled data: if uncertainty on unlabeled data is high we will enforce larger intra-class variance of seen classes to reduce the gap between variance of seen and novel classes, while if uncertainty is low we will enforce smaller intra-class variance on seen classes to encourage the model to fully exploit labeled data. In this way, using the uncertainty adaptive margin we control the intra-class variance of seen classes and ensure that discriminative representations for seen classes are not learned too fast compared to novel classes.

Given labeled instances $\mathcal{X}_l = \{x_i \in \mathbb{R}^N\}_{i=1}^n$ and unlabeled instances $\mathcal{X}_u = \{x_i \in \mathbb{R}^N\}_{i=1}^m$, ORCA first applies the embedding function $f_\theta : \mathbb{R}^N \to \mathbb{R}^D$ to obtain the feature representations $\mathcal{Z}_l = \{z_i \in \mathbb{R}^D\}_{i=1}^n$ and $\mathcal{Z}_u = \{z_i \in \mathbb{R}^D\}_{i=1}^m$ for labeled and unlabeled data, respectively. Here, $z_i = f_\theta(x_i)$ for every instance $x_i \in \mathcal{X}_l \cup \mathcal{X}_u$. On top of the backbone network, we add a classification head consisting of a single linear layer parameterized by a weight matrix $W : \mathbb{R}^D \to \mathbb{R}^{|\mathcal{C}_l \cup \mathcal{C}_u|}$, and followed by a softmax layer. Note that the number of classification heads is set to the number of previously seen classes and the expected number of novel classes. So, first $|\mathcal{C}_l|$ heads classify instances to one of the previously seen classes, while the remaining heads assign instances to novel classes. The final class/cluster prediction is calculated as $c_i = \mathrm{argmax}(W^T \cdot z_i) \in \mathbb{R}$. If $c_i \notin \mathcal{C}_l$, then $x_i$ belongs to novel classes. The number of novel classes $|\mathcal{C}_u|$ can be known and given as an input to the algorithm which is a typical assumption of clustering and novel class discovery methods. However, if the number of novel classes is not known ahead of time, we can initialize ORCA with a

large number of prediction heads/novel classes. The ORCA objective function then infers the number of classes by not assigning any instances to unneeded prediction heads so these heads never activate.

The objective function in ORCA combines three components (Figure 2) *(i)* supervised objective with uncertainty adaptive margin, *(ii)* pairwise objective, and *(iii)* regularization term:

$$\mathcal{L} = \mathcal{L}_S + \eta_1 \mathcal{L}_P + \eta_2 \mathcal{R}, \tag{1}$$

where $\mathcal{L}_S$ denotes supervised objective, $\mathcal{L}_P$ denotes pairwise objective and $\mathcal{R}$ is regularization. $\eta_1$ and $\eta_2$ are regularization parameters set to 1 in all our experiments. Pseudo-code of the algorithm is summarized in Algorithm 1 in Appendix B. We report sensitivity analysis to regularization parameters in Appendix C and next discuss the details of each objective term.

### 3.3 SUPERVISED OBJECTIVE WITH UNCERTAINTY ADAPTIVE MARGIN

First, the supervised objective with uncertainty adaptive margin forces the network to correctly assign instances to previously seen classes but controls the speed of learning this task in order to allow to simultaneously learn to form novel classes. We utilize the categorical annotations for the labeled data $\{y_i\}_{i=1}^n$ and optimize weights $W$ and backbone $\theta$. Categorical annotations can be exploited by using the standard cross-entropy (CE) loss as supervised objective:

$$\mathcal{L}_S = \frac{1}{n} \sum_{z_i \in \mathcal{Z}_l} - \log \frac{e^{W_{y_i}^T \cdot z_i}}{e^{W_{y_i}^T \cdot z_i} + \sum_{j \neq i} e^{W_{y_j}^T \cdot z_i}}, \tag{2}$$

However, using standard cross-entropy loss on labeled data creates an imbalance problem between the seen and novel classes, *i.e.*, the gradient is updated for seen classes $\mathcal{C}_s$, but not for novel classes $\mathcal{C}_n$. This can result in learning a classifier with larger magnitudes (Kang et al., 2019) for seen classes, leading the whole model to be biased towards the seen classes. To overcome the issue, we introduce an uncertainty adaptive margin mechanism and propose to normalize the logits as we describe next.

A key challenge is that seen classes are learned faster due to the supervised objective, and consequently they tend to have a smaller intra-class variance compared to the novel classes (Liu et al., 2020). The pairwise objective generates pseudo-labels for unlabeled data by ranking distances in the feature space, so the imbalance of intra-class variances among classes will result in error-prone pseudo-labels. In other words, instances from novel classes will be assigned to seen classes. To mitigate this bias, we propose to use an adaptive margin mechanism to reduce the gap between the intra-class variance of the seen and novel classes. Intuitively, at the beginning of the training, we want to enforce a larger negative margin to encourage a similarly large intra-class variance of the seen classes with respect to the novel classes. Close to the end of training, when clusters have been formed for the novel classes, we adjust the margin term to be nearly 0 so that labeled data can be fully exploited by the model, *i.e.*, the objective boils down to standard cross entropy defined in Eq. (2). We propose to capture intra-class variance using uncertainty. Thus, we adjust the margin using uncertainty estimate which achieves desired behavior — in early training epochs uncertainty is large which leads to a large margin, while as training proceeds the uncertainty becomes smaller which leads to a smaller margin.

Specifically, the supervised objective with uncertainty adaptive margin mechanism is defined as:

$$\mathcal{L}_S = \frac{1}{n} \sum_{z_i \in \mathcal{Z}_l} - \log \frac{e^{s(W_{y_i}^T \cdot z_i + \lambda \bar{u})}}{e^{s(W_{y_i}^T \cdot z_i + \lambda \bar{u})} + \sum_{j \neq i} e^{s W_{y_j}^T \cdot z_i}}, \tag{3}$$

where $\bar{u}$ is uncertainty and $\lambda$ is a regularizer defining its strength. Parameter $s$ is an additional scaling parameter that controls the temperature of the cross-entropy loss (Wang et al., 2018). To estimate uncertainty $\bar{u}$, we rely on the confidence of unlabeled instances computed from the output of the softmax function. In the binary setting, $\bar{u} = \frac{1}{|\mathcal{D}_u|} \sum_{x \in \mathcal{D}_u} \text{Var}(Y|X = x) = \frac{1}{|\mathcal{D}_u|} \sum_{x \in \mathcal{D}_u} \Pr(Y = 1|X) \cdot \Pr(Y = 0|X)$, which can be further approximated by:

$$\bar{u} = \frac{1}{|\mathcal{D}_u|} \sum_{x_i \in \mathcal{D}_u} 1 - \max_k \Pr(Y = k|X = x_i), \tag{4}$$

up to a factor of at most 2. Here, the $k$ goes over all classes. We use the same formula as an approximation for the group uncertainty in the multi-class setting, similar to (Cao et al., 2020b).

To adjust the margin properly, we need to constrain the magnitudes of the classifier since the unconstrained magnitudes of a classifier can negatively affect the tuning of the margin. To avoid the problem, we normalize the inputs and weights of the linear classifier, *i.e.,* $z_i = \frac{z_i}{|z_i|}$ and $W_j = \frac{W_j}{|W_j|}$.

## 3.4 PAIRWISE OBJECTIVE

The pairwise objective learns to predict similarities between pairs of instances such that the instances from the same class are grouped together. This part of the objective generates pseudo-labels for the unlabeled data to guide the training. By controlling intra-class variance of seen and novel classes using uncertainty adaptive margin, ORCA improves the quality of the pseudo-labels.

In particular, we transform the cluster learning problem into a pairwise similarity prediction task (Hsu et al., 2018; Chang et al., 2017). Given the labeled dataset $\mathcal{X}_l$ and unlabeled dataset $\mathcal{X}_u$, we aim to fine-tune our backbone $f_\theta$ and learn a similarity prediction function parameterized by a linear classifier $W$ such that instances from the same class are grouped together. To achieve this, we rely on the ground-truth annotations from the labeled set and pseudo-labels generated on the unlabeled set. Specifically, for the labeled set we already know which pairs should belong to the same class so we can use ground-truth labels. To obtain the pseudo-labels for the unlabeled set, we compute the cosine distance between all pairs of feature representations $z_i$ in a mini-batch. We then rank the computed distances and for each instance generate the pseudo-label for its most similar neighbor. Therefore, we only generate pseudo-labels from the most confident positive pairs for each instance within the mini-batch. For feature representations $\mathcal{Z}_l \cup \mathcal{Z}_u$ in a mini-batch, we denote its closest set as $\mathcal{Z}'_l \cup \mathcal{Z}'_u$. Note that $\mathcal{Z}'_l$ is always correct since it is generated using the ground-truth labels. The pairwise objective in ORCA is defined as a modified form of the binary cross-entropy loss (BCE):

$$\mathcal{L}_{\mathrm{P}} = \frac{1}{m+n} \sum_{\substack{z_i, z'_i \in \\ (\mathcal{Z}_l \cup \mathcal{Z}_u, \mathcal{Z}'_l \cup \mathcal{Z}'_u)}} - \log\langle \sigma(W^T \cdot z_i), \sigma(W^T \cdot z'_i) \rangle. \tag{5}$$

Here, $\sigma$ denotes the softmax function which assigns instances to one of the seen or novel classes. For labeled instances, we use ground truth annotations to compute the objective. For unlabeled instances, we compute the objective based on the generated pseudo-labels. We consider only the most confident pairs to generate pseudo-labels because we find that the increased noise in pseudo-labels is detrimental to cluster learning. Unlike (Hsu et al., 2018; Han et al., 2020; Chang et al., 2017) we consider only positive pairs as we find that including negative pairs in our objective does not benefit learning (negative pairs can be easily recognized). Our pairwise objective with only positive pairs is related to (Van Gansbeke et al., 2020). However, we update distances and positive pairs in an online fashion and thus benefit from improved feature representation during training.

## 3.5 REGULARIZATION TERM

Finally, the regularization term avoids a trivial solution of assigning all instances to the same class. In early stages of the training, the network could degenerate to a trivial solution in which all instances are assigned to a single class, *i.e.,* $|\mathcal{C}_u| = 1$. We discourage this solution by introducing a Kullback-Leibler (KL) divergence term that regularizes $\Pr(y|x \in \mathcal{D}_l \cup \mathcal{D}_u)$ to be close to a prior probability distribution $\mathcal{P}$ of labels $y$:

$$\mathcal{R} = KL\Big(\frac{1}{m+n} \sum_{z_i \in \mathcal{Z}_l \cup \mathcal{Z}_u} \sigma(W^T \cdot z_i) \,\big\|\, \mathcal{P}(y)\Big), \tag{6}$$

where $\sigma$ denotes softmax function. Since knowing prior distribution is a strong assumption in most applications, we regularize the model with maximum entropy regularization in all our experiments. Maximum entropy regularization has been used in pseudo-labeling based SSL (Arazo et al., 2020), deep clustering methods (Van Gansbeke et al., 2020) and training on noisy labels (Tanaka et al., 2018) to prevent the class distribution from being too flat. In the experiments, we show that this term does not negatively affect performance of ORCA even with unbalanced data distribution.

## 3.6 SELF-SUPERVISED PRETRAINING

We consider ORCA (as well as all the baselines) with and without self-supervised pretraining. On image datasets, we pretrain ORCA and baselines using self-supervised learning. Self-supervised

learning formulates a pretext/auxiliary task that does not need any manual curation and can be readily applied to labeled and unlabeled data. The pretext task guides the model towards learning meaningful representations in a fully unsupervised way. In particular, we rely on the SimCLR approach (Chen et al., 2020a). We pretrain the backbone $f_\theta$ on the whole dataset $\mathcal{D}_l \cup \mathcal{D}_u$ with a pretext objective. During training, we freeze the first layers of the backbone $f_\theta$ and update its last layers and classifier $W$. We adopt the same SimCLR pretraining protocol for all baselines. We also consider a setting without pretraining, where for the cell type annotation task, we do not do use any pretext task and ORCA starts from randomly initialized weights. Additionally, we report results with different pretraining strategies in Appendix C, including pretraining only on the labeled subset of the data $\mathcal{D}_l$ and replacing SimCLR with RotationNet (Kanezaki et al., 2018) .

## 4 EXPERIMENTS

### 4.1 EXPERIMENTAL SETUP

**Datasets.** We evaluate ORCA on four different datasets, including three standard benchmark image classification datasets CIFAR-10, CIFAR-100 (Krizhevsky, 2009) and ImageNet (Russakovsky et al., 2015), and a highly unbalanced single-cell Mouse Ageing Cell Atlas dataset from biology domain (Consortium et al., 2020). For single-cell dataset, we consider a realistic cross-tissue cell type annotation task where unlabeled data comes from different tissues compared to labeled data (Cao et al., 2020a) (details in Appendix B). For the ImageNet dataset, we sub-sample 100 classes following (Van Gansbeke et al., 2020). On all datasets, we use controllable ratios of unlabeled data and novel classes. We first divide classes into $50\%$ seen and $50\%$ novel classes. We then select $50\%$ of seen classes as the labeled dataset, and the rest as unlabeled set. We show results with different ratio of seen and novel classes and with $10\%$ labeled samples in the Appendix C.

**Baselines.** Given that the open-world SSL is a new setting no ready-to-use baselines exist. We thus extend novel class discovery, SSL and open-set recognition methods to the open-world SSL setting. Novel class discovery methods can not recognize seen classes, *i.e.,* match classes in unlabeled dataset to previously seen classes from the labeled dataset. We report their performance on novel classes, and extend these methods to be applicable to seen classes in the following way: We consider seen classes as novel (these methods effectively cluster the unlabeled data) and report performance on seen classes by using Hungarian algorithm to match some of the discovered classes with classes in the labeled data. We consider two methods: DTC (Han et al., 2019) and RankStats (Han et al., 2020).

On the other hand, traditional SSL and open-set recognition (OSR) methods can not discover novel classes. Thus we extend SSL and OSR methods to be applicable to novel classes in the following way: We use SSL/OSR to classify points into known classes and estimate out-of-distribution (OOD) samples. We report their performance on seen classes, and we then apply $K$-means clustering (Lloyd, 1982) to OOD samples to obtain clusters (novel classes). In this way we adapt two SSL methods to the open-world SSL setting: Deep Safe SSL (DS$^3$L) (Guo et al., 2020) and FixMatch (Sohn et al., 2020), and recent deep learning OSR method CGDL (Sun et al., 2020a). CGDL automatically rejects OOD samples. DS$^3$L considers novel classes in the unlabeled data by assigning low weights to OOD samples. To extend the method, we cluster samples with the lowest weights. For FixMatch, we estimate OOD samples based on softmax confidence scores. For both SSL methods, we use ground truth information of seen and novel classes partitions to determine the threshold for OOD samples.

On image datasets, we pretrain all novel class discovery and SSL baselines with SimCLR to ensure that the benefits of ORCA are not due to pretraining. The only exception is DTC which has its own specialized pretraining procedure on the labeled data (Han et al., 2019). As an additional baseline, we ran $K$-means clustering on representation obtained after SimCLR pretraining (Chen et al., 2020a). We also perform an extensive ablation studies to evaluate benefits of ORCA's approach. Specifically, we include comparison to a baseline in which adaptive margin cross-entropy loss in supervised objective is replaced with the standard cross-entropy loss, *i.e.,* zero margin (ZM) approach. Additionally, to evaluate effect of the adaptive margin we compare ORCA to a fixed negative margin (FNM). We find that margin value of $0.5$ achieves best performance (Appendix C) and we use that value in our experiments. We name the first baseline ORCA-ZM, and the second baseline ORCA-FNM. Additional implementation and experimental details can be found in Appendices.

Table 2: Mean accuracy computed over three runs. Asterisk (∗) denotes that the original method can not recognize seen classes (and we had to extend it). Dagger (†) denotes the original method can not detect novel classes (and we had to extend it). SimCLR and FixMatch are not applicable (NA) to the single-cell dataset. Improvement is computed as a relative improvement over the best baseline.

| Method | CIFAR-10 | | | CIFAR-100 | | | ImageNet-100 | | | Single-cell | | |
|---|---|---|---|---|---|---|---|---|---|---|---|---|
| | Seen | Novel | All | Seen | Novel | All | Seen | Novel | All | Seen | Novel | All |
| †FixMatch | 71.5 | 50.4† | 49.5 | 39.6 | 23.5† | 20.3 | 65.8 | 36.7† | 34.9 | NA | NA | NA |
| †DS³L | 77.6 | 45.3† | 40.2 | 55.1 | 23.7† | 24.0 | 71.2 | 32.5† | 30.8 | 76.2 | 29.7† | 26.4 |
| †CGDL | 72.3 | 44.6† | 39.7 | 49.3 | 22.5† | 23.5 | 67.3 | 33.8† | 31.9 | 74.1 | 30.4† | 25.6 |
| ∗DTC | 53.9∗ | 39.5 | 38.3 | 31.3∗ | 22.9 | 18.3 | 25.6∗ | 20.8 | 21.3 | 29.6∗ | 25.3 | 27.8 |
| ∗RankStats | 86.6∗ | 81.0 | 82.9 | 36.4∗ | 28.4 | 23.1 | 47.3∗ | 28.7 | 40.3 | 42.3∗ | 31.9 | 38.6 |
| ∗SimCLR | 58.3∗ | 63.4 | 51.7 | 28.6∗ | 21.1 | 22.3 | 39.5∗ | 35.7 | 36.9 | NA | NA | NA |
| ORCA-ZM | 87.6 | 86.6 | 86.9 | 55.2 | 32.0 | 34.8 | 80.4 | 43.7 | 55.1 | 89.5 | 35.1 | 47.6 |
| ORCA-FNM | 88.0 | 88.2 | 88.1 | 58.2 | 40.0 | 44.3 | 73.0 | 66.2 | 68.9 | 89.7 | 48.6 | 58.7 |
| **ORCA** | **88.2** | **90.4** | **89.7** | **66.9** | **43.0** | **48.1** | **89.1** | **72.1** | **77.8** | **89.9** | **65.2** | **72.9** |

**Remark.** We are aware that contrastive learning with perturbed input data can largely boost unsupervised learning on vision datasets (Van Gansbeke et al., 2020). We deliberately avoid using such tricks as they may not be easily transferrable to other domains. For readers who are interested, please feel free to add such tricks and reevaluate our model.

## 4.2 RESULTS

**Evaluation on benchmark datasets.** We report accuracy on seen and novel classes, as well as overall accuracy. Results in Table 2 show that ORCA consistently outperforms all baselines by a large margin. For example, on seen classes of the CIFAR-100 and ImageNet datasets ORCA achieves 21% and 25% improvements over the best baseline, respectively. On novel classes, ORCA outperforms baselines by 51% on the CIFAR-100, 96% on the ImageNet and 104% on the single-cell dataset. Furthermore, comparison of ORCA to ORCA-ZM and ORCA-FNM baselines clearly demonstrates the importance of introducing uncertainty adaptive margin for solving open-world SSL. Overall, our results demonstrate that *(i)* open-world SSL setting is hard and existing methods can not solve it adequately, and *(ii)* ORCA effectively addresses the challenges of the open-world SSL and achieves remarkable performance gains.

**Benefits of uncertainty adaptive margin.** We further systematically evaluate the effect of introducing uncertainty adaptive margin mechanism. On the CIFAR-100 dataset, we compare ORCA to ORCA-ZM and ORCA-FNM baselines during training (Figure 3). We report accuracy and uncertainty which captures intra-class variance, as defined in Eq. (4). At epoch 140, we decay the learning rate. Results show that ORCA-ZM is not able to reduce intra-class variance on novel classes during training, resulting in unsatisfactory performance on novel classes. On seen classes, ORCA-ZM reaches high performance very quickly, but its accuracy starts to decrease close to the end of training. The reason why learning rate triggers performance drop is that small learning rate can lead to overfitting issues (Li et al., 2019) which presents a problem in ORCA-ZM due to the difference in variances between seen and novel classes and noisy pseudolabels that deteriorate with small learning rate (Song et al., 2020). This shows that without uncertainty adaptive margin the model learns seen classes very quickly, but fails to achieve satisfying performance on novel classes. In contrast, ORCA effectively reduces intra-class variance on both seen and novel classes and consistently improves accuracy. This result is in perfect accordance with our key idea of slowly increasing discriminability of seen classes during training to ensure similar intra-class variance between seen and novel classes. Compared to ORCA-FNM, adaptive margin shows clear benefits on seen classes, achieving lower intra-class variance and better performance during the whole training process. In summary, negative margin ensures larger intra-class variance of seen classes allowing the model to learn to form novel classes, while adaptive margin ensures that the model can fully exploit labeled data as training proceeds.

Compared to other baselines in Table 2, ORCA outperforms their *final performance* after only 12 epochs. Additionally, in Appendix C we show that uncertainty adaptive margin improves the quality of pseudo-labels and demonstrate robustness to the uncertainty strength parameter $\lambda$. Taken together, our results strongly support the importance of the uncertainty adaptive margin.

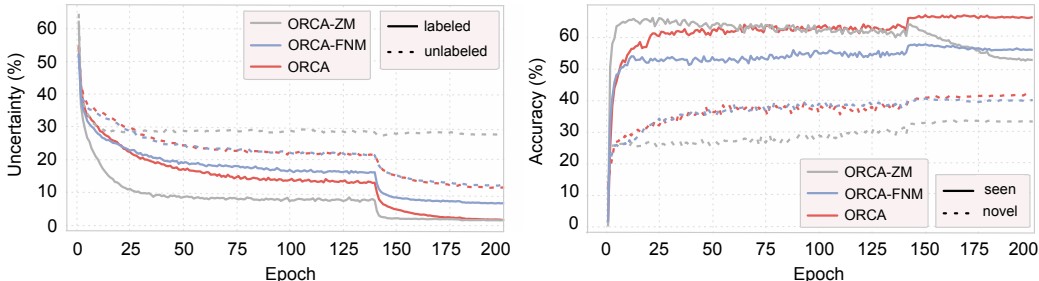

Figure 3: Effect of the uncertainty adaptive margin on the estimated uncertainty (left) and accuracy (right) during training on the CIFAR-100 dataset. At epoch $140$, we decay learning rate.

**Evaluation with the unknown number of novel classes.** ORCA and other baselines assume that number of novel classes is known. However, in practice we often do not know number of classes in advance. In such case, we can apply ORCA by first estimating the number of classes. To evaluate performance on the CIFAR-100 dataset which has $100$ classes, we first estimate the number of clusters using technique proposed in (Han et al., 2019) to be $124$. We then use the estimated number of classes to re-test all the algorithms. ORCA automatically prunes number of classes by not utilizing all initialized classification heads, and finds $114$ novel clusters. Results in Table 3 show that ORCA outperforms novel class discovery baselines, achieving $97\%$ improvement over RankStats. Moreover, with the estimated number of classes ORCA achieves only slightly worse results compared to the setting in which the number of classes is known a priori. We further analyzed the $14$ additional heads used in ORCA unassigned with Hungarian algorithm and found that they are related to small clusters, *i.e.*, the average number of samples in the unassigned heads is only $16$. The NMI on these classes is $59.4$, which is slightly higher than the NMI on the assigned heads. This indicates that additional clusters contain smaller subclasses of the correct class and belong to meaningful clusters. We perform additional ablation study with large number of classes in Appendix C.

**Ablation study on the objective function.** The objective function in ORCA consists of supervised objective with uncertainty adaptive margin, pairwise objective, and regularization term. To investigate importance of each part, we conduct an ablation study in which we modify ORCA by removing: *(i)* supervised objective (*i.e.,* w/o $\mathcal{L}_S$), and *(ii)* regularization term (*i.e.,* w/o $\mathcal{R}$). In the first case, we rely only on the regularized pairwise objective to solve the problem, while in the latter case we use unregularized supervised and pairwise objectives. We note that the pairwise objective is required to be able to discover novel classes. The results shown in Table 4 on the CIFAR-100 dataset demonstrate that both supervised objective $\mathcal{L}_S$ and regularization $\mathcal{R}$ are essential parts of the objective function. Additional experimental results with unbalanced data distribution are reported in the Appendix C.

Table 3: Mean accuracy and normalized mutual information (NMI) on CIFAR-100 dataset over three runs with unknown number of novel classes.

| Method | Seen | Novel | Novel (NMI) | All |
|---|---|---|---|---|
| DTC | 30.7* | 15.4 | 33.7 | 14.5 |
| RankStats | 33.7* | 22.1 | 37.4 | 20.3 |
| ORCA | 66.3 | 40.0 | 50.9 | 46.4 |

Table 4: Ablation study on the components of the objective function on the CIFAR-100 dataset. We report mean accuracy and NMI over three runs.

| Approach | Seen | Novel | Novel (NMI) | All |
|---|---|---|---|---|
| w/o $\mathcal{L}_S$ | 12.2 | 13.4 | 25.4 | 10.0 |
| w/o $\mathcal{R}$ | 63.6 | 25.9 | 44.6 | 29.7 |
| ORCA | 66.9 | 43.0 | 52.1 | 48.1 |

## 5 CONCLUSION

We introduced open-world SSL setting in which novel classes can appear in the unlabeled test data and the model needs to assign instances either to classes seen in the labeled data, or form novel classes and assign instances to them. To address the problem, we proposed ORCA, a method based on the uncertainty adaptive margin mechanism which controls intra-class variance of seen and novel classes during training. Our extensive experiments show that ORCA effectively solves open-world SSL and outperforms alternative baselines by a large margin. Our work advocates for a shift from traditional closed-world setting to the more realistic open-world evaluation of machine learning models.

## REPRODUCIBILITY STATEMENT

The pseudo-code of the algorithm and implementation details are provided in the Appendix C. The code of ORCA is publicly available at https://github.com/snap-stanford/orca.

## ACKNOWLEDGEMENTS

The authors thank Kexin Huang, Hongyu Ren, Yusuf Roohani, Camilo Ruiz, Pranay Reddy Samala, Tailin Wu and Michael Zhang for their feedback on our manuscript. We also gratefully acknowledge the support of DARPA under Nos. HR00112190039 (TAMI), N660011924033 (MCS); ARO under Nos. W911NF-16-1-0342 (MURI), W911NF-16-1-0171 (DURIP); NSF under Nos. OAC-1835598 (CINES), OAC-1934578 (HDR), CCF-1918940 (Expeditions), IIS-2030477 (RAPID), NIH under No. R56LM013365; Stanford Data Science Initiative, Wu Tsai Neurosciences Institute, Chan Zuckerberg Biohub, Amazon, JPMorgan Chase, Docomo, Hitachi, Intel, KDDI, Toshiba, NEC, and UnitedHealth Group. J. L. is a Chan Zuckerberg Biohub investigator.

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

## A    ADDITIONAL RELATED WORK

**Universal domain adaptation (UniDA).** UniDA (You et al., 2019; Saito et al., 2020) does not impose any assumptions on the overlap between classes of source and target datasets. However, all novel classes that appear in the target dataset are considered as "unknowns" and the goal is to reject them like in the open-set recognition and robust SSL. As a domain adaption setting, UniDA considers feature distribution shift between source and target datasets. Open-world SSL is a generalization of the SSL setting so it does not assume a feature distribution shift.

**Margin loss.** Based on the insight that margin term in cross-entropy loss can adjust intra- and inter-class variations, losses like large-margin softmax (Liu et al., 2016), angular softmax (Liu et al., 2017), circle loss (Sun et al., 2020b) and additive margin softmax (Wang et al., 2018) have been proposed to achieve better classification accuracy. Cao et al. (2019) assigned class-specific margins to encourage the optimal trade-off in generalization between frequent and rare classes. Liu et al. (2020) used negative margin to enlarge intra-class variance and reduce inter-class variance, leading to better performance on novel classes in the few-shot learning setting.

## B    IMPLEMENTATION DETAILS

Our core algorithm is developed using PyTorch (Paszke et al., 2019) and we conduct all the experiments with NVIDIA RTX 2080 Ti. Note that our proposed method ORCA adds no computational overhead over the previous methods that are comparable, *e.g.*, DTC and RankStats. Empirically, our experiments on CIFAR take less than an hour and the experiments on the ImageNet take a few hours.

**Evaluation metrics.** To measure performance on unlabeled data, we follow the evaluation protocol in novel class discovery (Han et al., 2019; 2020). On seen classes, we report accuracy. On novel classes, we report accuracy and normalized mutual information (NMI). To compute accuracy on the novel classes, we first solve optimal assignment problem using Hungarian algorithm (Kuhn, 1955). When reporting accuracy on all classes jointly, we solve optimal assignment using both seen and novel classes.

**Implementation details for the CIFAR datasets.** We follow the simple data augmentation suggested in (He et al., 2016) with only random crop and horizontal flip. We use a modified ResNet-18 that is compatible with input size $32 \times 32$ following (Han et al., 2020) and repeat all experiments for three runs. We train the model using standard SGD with a momentum of 0.9 and a weight decay of $5 \times 10^{-4}$. The model is trained for 200 epochs with a batch size of 512. We anneal the learning rate by a factor of 10 at epoch 140 and 180. Similar to (Han et al., 2020), we only update the parameters of the last block of ResNet in the second training stage to avoid over-fitting. We set hyperparameters to the following default values: $s = 10, \lambda = 1, \eta_1 = 1, \eta_2 = 1$. Temperature $s$ scales up the dot product and allows the value after softmax to be able to reach 1. We set it to 10 which is a common choice (Wang et al., 2018), but it is not a sensitive parameter. All hyperparameters remain the same across all experiments unless otherwise specified. We show robustness to the parameters in Appendix C.

**Implementation details for the ImageNet dataset.** We follow the standard data augmentation including random resized crop and horizontal flip (He et al., 2016). We use ResNet-50 as the backbone. We train the model using standard SGD with a momentum of 0.9 and a weight decay of $1 \times 10^{-4}$. The model is trained for 90 epochs with a batch size of 512. We anneal the learning rate by a factor of 10 at epoch 30 and 60. Similar to (Han et al., 2020), we only update the parameters of the last block of the ResNet in the second training stage to avoid over-fitting. We set hyperparameters to the following default values: $s = 10, \lambda = 1, \eta_1 = 1, \eta_2 = 1$. They remain the same across all experiments unless otherwise specified.

**Implementation details for the single-cell dataset.** Single-cell dataset is a highly unbalanced real-world dataset (skewness=3.8). We consider cross-tissue cell type annotation task (Cao et al., 2020a) and take cell types with at least 400 examples per class. The dataset consists of $93,718$ cells from 50 cell types collected across 23 organs of the mouse model organism. The features correspond to $2,866$ highly variable genes. The largest class has $13,268$ examples, while the smallest class has 479

examples. We do not use any pretraining nor data augmentation for single-cell dataset. We use a simple backbone network structure containing two fully-connected layers with batch normalization, ReLu activation and dropout. We use Adam optimizer with an initial learning rate of $10^{-3}$ and a weight decay $0$. The model is trained with a batch size of $512$ for 20 epochs. We set hyperparameters to the following default values: $s = 10$, $\lambda = 1$, $\eta_1 = 1$, $\eta_2 = 1$. They remain the same across all experiments unless otherwise specified.

**Implementation details for baselines.** We compare ORCA to methods from three settings: *(i)* novel class discovery methods, *(ii)* semi-supervised methods, *(iii)* open-set recognition methods. First, novel class discovery methods have the ability to discover novel classes but lack the ability to recognize seen classes. Therefore, we apply them to discover novel classes and then evaluate performance by matching discovered classes/clusters to ground truth labels by solving optimal assignment using Hungarian algorithm (Kuhn, 1955). Notice that we are giving an advantage to these methods compared to ORCA since we are using ground truth information for matching classes and these method have never identified seen classes. Second, to extend semi-supervised methods we first estimate out-of-distribution (OOD) samples that should belong to novel classes and then cluster OOD samples to assign OOD samples to different classes. For FixMatch which is a standard semi-supervised method, we first need to estimate OOD samples. We use the confidence scores from the softmax to estimate OOD samples, *i.e.*, lower confidence means lower probability that the example belongs to the seen class. On the other hand, DS3L already assumes that OOD samples are present in the data and assigns low weights to these examples so we can directly use the weights to estimate examples from novel classes. However, in both cases we need to determine a threshold for what is considered to be an OOD example. To determine a threshold, we use ground truth information, *i.e.*, the ratio of samples belonging to seen and novel classes. We set the threshold so that the ratio of seen and unseen in the prediction aligns with the ground truth split ($50\%$ in our experiments). After thresholding OOD examples, we can simply cluster them by running $K$-means clustering. Finally, open-set recognition methods automatically reject OOD samples and we cluster OOD samples like in semi-supevised methods.

**ORCA algorithm.** We summarize the steps of ORCA in Algorithm 1.

---

**Algorithm 1** ORCA: Open-woRld with unCertainty based Adaptive margin

---

**Require:** Labeled subset $\mathcal{D}_l = \{(x_i, y_i)\}_{i=1}^n$, unlabeled subset $\mathcal{D}_u = \{(x_i)\}_{i=1}^m$, expected number of novel classes, a parameterized backbone $f_\theta$, linear classifier with weight $W$.
1: Pretrain the model parameters $\theta$ with pretext loss
2: **for** epoch $= 1$ to $E$ **do**
3:   $\bar{u} \leftarrow \text{EstimateUncertainty}(\mathcal{D}_u)$
4:   **for** $t = 1$ to $T$ **do**
5:    $\mathcal{X}_l, \mathcal{X}_u \leftarrow \text{SampleMiniBatch}(\mathcal{D}_l \cup \mathcal{D}_u)$
6:    $\mathcal{Z}_l, \mathcal{Z}_u \leftarrow \text{Forward}(\mathcal{X}_l \cup \mathcal{X}_u; f_\theta)$
7:    $\mathcal{Z}'_l, \mathcal{Z}'_u \leftarrow \text{FindClosest}(\mathcal{Z}_l \cup \mathcal{Z}_u)$
8:    Compute $\mathcal{L}_P$ using (5)
9:    Compute $\mathcal{L}_S$ using (3)
10:    Compute $\mathcal{R}$ using (6)
11:    $f_\theta \leftarrow \text{SGD with loss } \mathcal{L}_{\text{BCE}} + \eta_1 \mathcal{L}_{\text{CE}} + \eta_2 \mathcal{L}_{\text{R}}$
12:   **end for**
13: **end for**

---

# C ADDITIONAL RESULTS

**Results with the reduced number of labeled data.** Instead of constructing labeled set with $50\%$ examples labeled in seen classes, we evaluate the performance of ORCA and baselines on the labeled set with only $10\%$ labeled examples. Results on all four benchmark datasets are shown in Table 5. We find that ORCA's substantial improvements over baselines are retained.

**Effect of the number of novel classes.** We next systematically evaluate performance when varying the ratio of seen and novel classes in the unlabeled set on the CIFAR-100 dataset (Figure 4). On seen

Table 5: Mean accuracy on benchmark datasets computed over three runs. For each seen class, we only label $10\%$ examples. On seen classes, asterisk ($*$) denotes that the original method by itself can not recognize seen classes, and we extend it by matching discovered clusters to classes in the labeled data. On novel classes, dagger ($\dagger$) denotes the original method cannot detect novel classes and we extend it by performing clustering over out-of-distribution samples.

| Method | CIFAR-10 | | | CIFAR-100 | | | ImageNet-100 | | | Single-cell | | |
| | Seen | Novel | All | Seen | Novel | All | Seen | Novel | All | Seen | Novel | All |
|---|---|---|---|---|---|---|---|---|---|---|---|---|
| FixMatch | 64.3 | 49.4$^\dagger$ | 47.3 | 30.9 | 18.5$^\dagger$ | 15.3 | 60.9 | 33.7$^\dagger$ | 30.2 | NA | NA | NA |
| DS$^3$L | 70.5 | 46.6$^\dagger$ | 43.5 | 33.7 | 15.8$^\dagger$ | 15.1 | 64.3 | 28.1$^\dagger$ | 25.9 | 70.8 | 26.4$^\dagger$ | 21.8 |
| DTC | 42.7* | 31.8 | 32.4 | 22.1* | 10.5 | 13.7 | 24.5* | 17.8 | 19.3 | 25.8* | 20.2 | 22.4 |
| RankStats | 71.4* | 63.9 | 66.7 | 20.4* | 16.7 | 17.8 | 41.2* | 26.8 | 37.4 | 40.1* | 27.6 | 35.3 |
| ORCA-ZM | 82.7 | 70.6 | 72.4 | 35.8 | 23.9 | 22.2 | 78.9 | 41.4 | 50.3 | 79.9 | 31.8 | 41.7 |
| ORCA-FNM | 77.9 | 71.8 | 74.8 | 32.5 | 30.8 | 31.6 | 66.1 | 57.7 | 62.3 | 79.4 | 40.3 | 45.3 |
| ORCA | **82.8** | **85.5** | **84.1** | **52.5** | **31.8** | **38.6** | **83.9** | **60.5** | **69.7** | **80.6** | **52.8** | **61.3** |

classes, we find that ORCA's performance is stable across all values with slightly higher performance as the percentage of labeled data increases. On novel classes, ORCA expectedly achieves lower performance with more unlabeled data. However, remarkably even with the $90\%$ of novel classes, ORCA achieves higher performance than all the baselines in Table 2 with $50\%$ of novel classes.

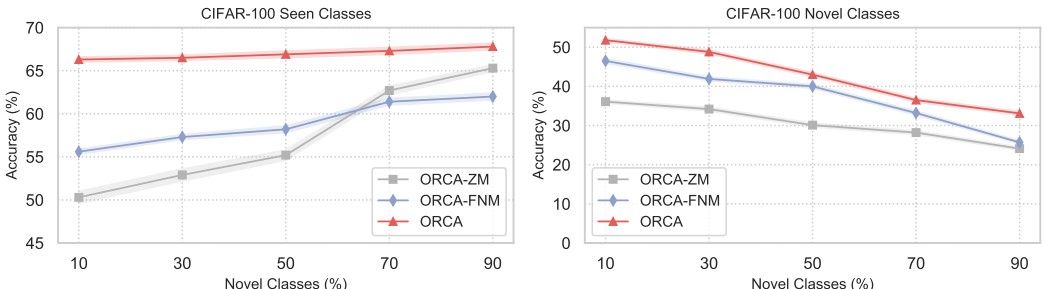

Figure 4: Mean accuracy on recognizing seen classes when varying percentage of seen/novel classes on the CIFAR-100 dataset. Error bands represent standard deviation across different runs.

**Fixed negative margin results.** We experiment with the different values of the margin threshold and we find that the margin value of $0.5$ achieves best performance. Results on the CIFAR-100 dataset are shown in Table 6. We use this value in all our experiments with fixed negative margin (ORCA-FNM).

Table 6: Accuracy with different fixed margin value on the CIFAR-100 dataset.

| $\lambda$ | Seen | Novel | All |
|---|---|---|---|
| 0.3 | 62.4 | 35.7 | 40.5 |
| 0.4 | 60.8 | 37.9 | 43.2 |
| 0.5 | 58.2 | 40.0 | 44.3 |
| 0.6 | 48.1 | 41.7 | 43.0 |
| 0.7 | 42.9 | 42.2 | 42.6 |

**Sensitivity analysis of $\eta_1$ and $\eta_2$.** Parameters $\eta_1$ and $\eta_2$ define importance of the supervised objective and maximum entropy regularization, respectively. To analyze their effect on the performance, we vary these parameters and evaluate ORCA's performance on the CIFAR-100 dataset. Results are shown in Table 7. We find that higher values of $\eta_1$ achieve slightly better performance on seen classes. This result agrees well with the intuition: giving more importance to the supervised objective improves performance on the seen classes. The effect of parameter $\eta_2$ on seen classes is opposite and lower values of $\eta_2$ achieve better performance on seen classes. On novel classes, the optimal performance is obtained when $\eta_1$ and $\eta_2$ are set to 1.

Table 7: Mean accuracy computed over three runs with different values of $\eta_1$ and $\eta_2$ on the CIFAR-100 dataset with $50\%$, $50\%$ split for seen and novel classes.

| $\eta_1$ | Seen | Novel | All | $\eta_2$ | Seen | Novel | All |
|---|---|---|---|---|---|---|---|
| 0.6 | 65.7 | 42.3 | 47.5 | 0.6 | 71.4 | 28.0 | 30.2 |
| 0.8 | 66.0 | 41.9 | 47.1 | 0.8 | 68.5 | 39.3 | 43.0 |
| 1.0 | 66.9 | 43.0 | 48.1 | 1.0 | 66.9 | 43.0 | 48.1 |
| 1.2 | 66.9 | 42.7 | 47.6 | 1.2 | 66.7 | 42.8 | 47.9 |
| 1.4 | 66.6 | 41.9 | 46.8 | 1.4 | 66.3 | 41.8 | 47.7 |

**Sensitivity analysis of uncertainty regularizer $\lambda$.** The intention of introducing the uncertainty adaptive margin is to enforce the group of labeled and unlabeled data to have similarly large intra-class variances. Here we inspect how does the uncertainty regularizer $\lambda$ affect performance. The results are shown in Table 8. A slightly larger $\lambda$ achieves higher accuracy on the novel classes with the cost of lower accuracy on seen classes. In contrast, smaller values of $\lambda$ achieve slightly better performance on seen classes. In general, ORCA is very robust to the selection of the parameter.

Table 8: Mean accuracy over three runs with different values of regularizer $\lambda$ on CIFAR-100 dataset with $50\%$, $50\%$ split for seen and novel classes.

| $\lambda$ | Seen | Novel | All |
|---|---|---|---|
| 0.6 | 67.0 | 42.6 | 47.5 |
| 0.8 | 67.0 | 42.8 | 47.6 |
| 1.0 | 66.9 | 43.0 | 48.1 |
| 1.2 | 66.6 | 43.4 | 48.0 |
| 1.4 | 66.0 | 43.5 | 48.2 |

**Benefits of uncertainty adaptive margin on the quality of pseudo-labels.** The benefit of the uncertainty adaptive margin is that it reduces the bias towards seen classes. To evaluate the effect of uncertainty adaptive margin on the quality of generated pseudo-labels during training, we compare the accuracy of adaptive margin to baseline approach with zero margin (ZM) and fixed negative margin (FNM) approach on the CIFAR-100 dataset. We report accuracy of generated pseudo-labels in Figure 5, following the same setting as in Figure 3. This analysis additionally confirms that adaptive margin significantly increases the accuracy of the estimated pseudo-labels.

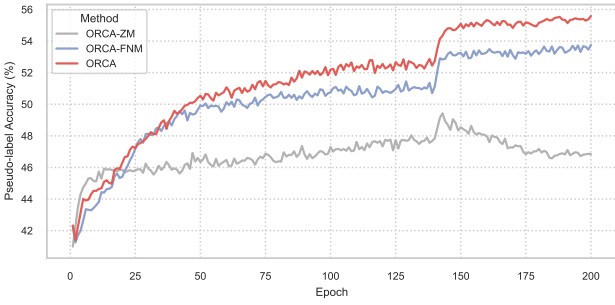

Figure 5: Effect of the uncertainty adaptive margin on the quality of estimated pseudo-labels during training on the CIFAR-100 dataset.

**Results with the unbalanced data distribution.** We further analyze whether the maximum entropy regularization negatively affects performance when the distribution of the classes is unbalanced (Table 9). To test performance with unbalanced classes, we artificially make class distributions in CIFAR-10 and CIFAR-100 datasets long-tailed by following an exponential decay in sample sizes across different classes. The imbalance ratio between sample sizes of the most frequent and least frequent class is set to 10 in the experiments. We find that the proposed regularization consistently improves performance over non-regularized model even with unbalanced distribution.

Table 9: Mean accuracy and normalized mutual information (NMI) on the unbalanced CIFAR-10 and CIFAR-100 datasets calculated over three runs.

| | | | CIFAR-10 | | | | CIFAR-100 | |
|---|---|---|---|---|---|---|---|---|
| Approach | Seen | Novel | Novel (NMI) | All | Seen | Novel | Novel (NMI) | All |
| w/o $\mathcal{R}$ | 84.2 | 61.4 | 64.6 | 62.8 | 55.6 | 35.4 | 50.6 | 35.2 |
| w/ $\mathcal{R}$ | 90.4 | 82.9 | 74.6 | 69.0 | 65.0 | 37.2 | 53.8 | 40.5 |

**Results with pretraining on the labeled data.** On image datasets, we pretrain ORCA and all other baselines jointly on labeled and unlabeled data. Here we analyze how does pretraining only on the labeled subset of the data affect performance. The results are shown in Table 10. Compared to pretraining on the whole dataset, we find that ORCA only slightly degrades performance when pretrained only on the labeled subset of the data. This additionally confirms that the major contribution in ORCA's high performance on novel classes lies in its objective function and not in the pretraining strategy.

Table 10: Comparison of pretraining over both labeled and unlabeled data vs. over the labeled data only on the CIFAR-100 dataset. We report mean accuracy over three runs.

| Method | Pretrain on all | | | Pretrain on labeled | | |
|---|---|---|---|---|---|---|
| | Seen | Novel | All | Seen | Novel | All |
| ORCA-ZM | 55.2 | 32.0 | 34.8 | 51.0 | 27.2 | 29.4 |
| ORCA-FNM | 58.2 | 40.0 | 44.3 | 56.7 | 35.4 | 41.2 |
| ORCA | 66.9 | 43.0 | 48.1 | 66.4 | 40.1 | 45.2 |

**Results with different pretraining strategies.** On image datasets, we pretrain ORCA using SimCLR (Chen et al., 2020a). Here we experiment with RotationNet (Kanezaki et al., 2018) as a pretraining strategy. Results shown in Table 11 demonstrate that further benefits of ORCA can be expected using RotationNet as the pretraining strategy.

Table 11: Comparison of pretraining strategies on the CIFAR-100 dataset. We report mean accuracy over three runs.

| Method | SimCLR | | | RotationNet | | |
|---|---|---|---|---|---|---|
| | Seen | Novel | All | Seen | Novel | All |
| ORCA-ZM | 55.2 | 32.0 | 34.8 | 57.6 | 33.4 | 37.9 |
| ORCA-FNM | 58.2 | 40.0 | 44.3 | 59.2 | 42.7 | 47.3 |
| ORCA | 66.9 | 43.0 | 48.1 | 70.0 | 45.6 | 51.4 |

**Results with different margin estimation approaches.** We additionally experimented with *(i)* using entropy with proper normalization for margin value, and *(ii)* using linear margin scheduling instead of uncertainty based adaptive margin. The results on the CIFAR-100 dataset are shown in Table 12. The results show that both approaches result in suboptimal performance compared to uncertainty adaptive margin approach.

**Results with different number of novel classes.** ORCA can infer number of novel classes by not activating some classification heads. To systematically validate this, we performed an ablation study by initializing ORCA with different number of classes. Results on the CIFAR-100 dataset are shown in Table 13. The results show that even with initially high number of classes ORCA retains high performance.

Table 12: Results with different margin estimation approaches on the CIFAR-100 dataset. We report mean accuracy over three runs.

| Approach | Seen | Novel | All |
|---|---|---|---|
| Entropy | 66.5 | 41.7 | 46.5 |
| Linear schedule | 65.1 | 40.7 | 45.9 |
| ORCA | 66.9 | 43.0 | 48.1 |

Table 13: Results with different number of novel classes on the CIFAR-100 dataset. We report mean accuracy over three runs.

| Initial number of novel classes | Seen | Novel | All |
|---|---|---|---|
| 100 | 66.9 | 43.0 | 48.1 |
| 200 | 66.5 | 41.7 | 47.2 |
| 400 | 65.3 | 39.6 | 43.4 |

