# OpenReview forum: "Open-World Semi-Supervised Learning"
_ICLR.cc/2022/Conference — ICLR 2022 Poster_

### Official Review · Reviewer_NjL4 · 2021-11-01

**Correctness:** 4
**Technical Novelty And Significance:** 2
**Empirical Novelty And Significance:** 3
**Recommendation:** 6
**Confidence:** 4

**Main Review:**

**Strengths**

- This work introduces a realistic semi-supervised problem that accounts for the presence of samples from novel classes in the unlabeled set. This problem setup is interesting and would make the developed SSL solutions applicable to more practical scenarios.
- The proposed solution achieves promising results. Even though the technical novelty of different components of the proposed solution is not high, incorporating these multiple components to solve different aspects of the challenging open-world semi-supervised learning problem is novel enough.
- The paper is well written and easy to follow.
- The ablation is extensive. Besides, the work includes analysis under various challenging conditions like class imbalance, varying percentage of seen and novel classes, etc.

**Weakness and Concerns**

- The effectiveness of the pairwise objective should greatly depend on the quality of self-supervised features. The proposed method and most of the baselines have utilized SimCLR pretrained weights. An analysis would have been nice to see how the performance of the proposed method changes with different self-supervised pretraining schemes; especially the ones with pretext tasks like RotNet, Jigsaw etc. Besides, for a lot of datasets the self-supervised pretrained weights might not be available; for instance, the single-cell dataset. Therefore, it is crucial to know how the method performs without incorporating the self-supervised pretrained weights on the benchmark datasets: CIFAR-10, CIFAR-100, ImageNet-100.
- The experiments on the main text used a large portion of labeled data (50%), which is not a very practical setup for semi-supervised learning. However, the appendix includes results with 10% labeled data. Since the SOTA closed-world SSL methods are very label efficient, similar experiments (with 1%, 5%, etc labeled data) could have been conducted.
- Pseudo-labels for pairwise objective have been generated by finding the most confident pairs. However, for datasets with a large number of classes, more than one sample might not be present in a minibatch if the batch-size is not sufficiently large. Therefore, it would be interesting to know the sensitivity of the proposed method to the batch-size parameter.
- How were the hyperparameters for the proposed method tuned? Did the authors use a validation set for this purpose? If that is the case, then did that validation set contain labeled samples from novel classes? Because using labeled samples from novel classes to tune hyperparameters would make the proposed method less practical.
- Comparison with an unsupervised clustering method like SCAN[1] would have been interesting.

**Questions and Comments**

- The results reported in Table 2, and 5 demonstrates that the accuracy on novel classes is higher than that of seen classes for the CIFAR-10 dataset. Since for all the other methods the seen class performance is higher it would be interesting to know any insight behind this phenomenon.
- In parallel to the zero margin and fixed margin experiments have the authors conducted any experiments by decaying the margin for CE loss with a predefined schedule (linear, sigmoid, cosine, etc)? Even though this approach involves some hyperparameter tuning it would reduce the computation needed to compute the uncertainty adaptive margin.

[1] Gansbeke, et al. SCAN: Learning to Classify Images without Labels. ECCV 2020.

**Summary Of The Paper:**

The paper introduces a new semi-supervised learning problem, open-world semi-supervised learning, where the objective is to recognize samples from seen classes as well as cluster samples from novel classes. The proposed method consists of three components to address the different aspects of this problem:  a supervised CE loss with uncertainty adaptive margin to recognize samples from seen classes, a pairwise objective to cluster samples from novel classes, and a regularization term to avoid degenerate solutions. The effectiveness of the proposed method has been validated on multiple benchmark datasets.

**Summary Of The Review:**

This work introduces a new semi-supervised learning problem that has more real-world applications. Even though the technical novelty of different components of the proposed method is not high, the solution is well motivated and achieves promising performance on multiple datasets. However, some of the points are not convincing and require further clarification to assess the significance of the work. I would appreciate it if the authors address these concerns in their response.

---

> ### Author Response · Authors · 2021-11-17
> **Response to Reviewer NjL4 (part 1)**
>
> We thank the reviewer for the insightful suggestions and positive evaluation of our work. We are glad that the reviewer appreciates the practical usefulness of our setting, the novelty of our method, extensive ablation studies and thinks that the paper is well written. We provide a detailed response below and we hope that our response helps in increasing the reviewer's confidence.
>
> **RE: The effectiveness of the pairwise objective should greatly depend on the quality of self-supervised features. The proposed method and most of the baselines have utilized SimCLR pretrained weights. An analysis would have been nice to see how the performance of the proposed method changes with different self-supervised pretraining schemes ...**
>
> We thank the reviewer for the suggestion. We experimented with switching the self-supervised algorithm to RotationNet. We find that additional improvements can be achieved using RotationNet. Table on the CIFAR-100 dataset is shown below. In response to the reviewers feedback, we included the result in the Appendix (Table 11).
>
> | SSL         | Method   | Seen | Novel | All  |
> |-------------|----------|------|-------|------|
> | SimCLR      | ORCA-ZM  | 55.2 | 32.0  | 34.8 |
> | SimCLR      | ORCA-FNM | 58.2 | 40.0  | 44.3 |
> | SimCLR      | ORCA     | 66.9 | 43.0  | 48.1 |
> | RotationNet | ORCA-ZM  | 57.6 | 33.4  | 37.9 |
> | RotationNet | ORCA-FNM | 59.2 | 42.7  | 47.3 |
> | RotationNet | ORCA     | 70.0 | 45.6  | 51.4 |
>
> We agree with the reviewer that there are datasets on which it is not possible to adopt self-supervised pretraining. For that reason, we already have an experiment in our paper on a single-cell dataset without any pretraining. In that experiment, all methods are initialized randomly. Results show that ORCA achieves high performance and achieves 88% improvement over best baseline.
>
> **RE: The experiments on the main text used a large portion of labeled data (50%), which is not a very practical setup for semi-supervised learning. However, the appendix includes results with 10% labeled data. Since the SOTA closed-world SSL methods are very label efficient, similar experiments (with 1%, 5%, etc labeled data) could have been conducted.**
>
> We thank the reviewer for the question. We ran ORCA with 5% labeled examples on the CIFAR-100 dataset and found similar performance as with 10% labeled examples. We report ORCA’s performance below. Remarkably, even with 5% labeled examples, ORCA outperforms all baselines when they use 10% of labeled examples. Thus, ORCA is applicable with less labeled examples and we showed in our work that it achieves even higher performance gains with less labeled data (10% in Table 5 compared to 50% in Table 2).
>
> | Labeled Percent  | Seen | Novel | All  |
> |-------------|------|-------|------|
> | 5%     | 49.4 | 30.5  | 35.7 |
> | 10%   |  52.5 | 31.8 | 38.6 |
>
> **RE: Pseudo-labels for pairwise objective have been generated by finding the most confident pairs. However, for datasets with a large number of classes, more than one sample might not be present in a minibatch if the batch-size is not sufficiently large. Therefore, it would be interesting to know the sensitivity of the proposed method to the batch-size parameter.**
>
> We thank the reviewer for the insightful question. Intuitively, larger batch size should improve performance of ORCA as reviewer observed. Additionally, we empirically confirm that indeed larger batch size improves performance of ORCA. The results are shown in the table below on the CIFAR-100 dataset. To deal with the larger number of classes, ORCA can be easily extended by accumulating several batches when computing positive candidates.
>
> | Batch Size  | Seen | Novel | All  |
> |-------------|------|-------|------|
> | 256     | 66.8 | 35.7  | 40.9 |
> | 512      | 66.9 | 43.0  | 48.1 |
>
> **RE: How were the hyperparameters for the proposed method tuned? Did the authors use a validation set for this purpose? If that is the case, then did that validation set contain labeled samples from novel classes? Because using labeled samples from novel classes to tune hyperparameters would make the proposed method less practical.**
>
> We agree with the reviewer that there’s no way to tune the hyperparameters for real world deployment. Thus, for method specific parameters, we use the same hyperparameters across all datasets and we did not tune the hyperparameters. In particular, we set \lambda, \eta_1, \eta_2 to 1 in all our experiments (please see Appendix A). Additionally, in the Appendix we show robustness of ORCA to these parameters and give intuition how to set them (Tables 6-8).
>
> Regarding other parameters, we use standard backbones (e.g., ResNet-50, ResNet-18) and follow the training procedure of papers cited in the Appendix. We did not tune any of the hyperparameters.

---

> > ### Author Response · Authors · 2021-11-17
> > **Response to Reviewer NjL4 (part 2)**
> >
> > **RE: Comparison with an unsupervised clustering method like SCAN[1] would have been interesting.**
> >
> > We thank the reviewer for the question. We have benchmarked SCAN on the CIFAR-100 dataset as well. SCAN[1] fails to utilize the labeled information and expectedly leads to significantly worse performance.
> >
> > | Method  | Seen | Novel | All  |
> > |-------------|------|-------|------|
> > | SCAN     | 27.2* | 23.3  | 24.6 |
> > | ORCA      | 66.9 | 43.0  | 48.1 |
> >
> > **RE: The results reported in Table 2, and 5 demonstrates that the accuracy on novel classes is higher than that of seen classes for the CIFAR-10 dataset. Since for all the other methods the seen class performance is higher it would be interesting to know any insight behind this phenomenon.**
> >
> > We thank the reviewer for the insightful question. The higher per-class accuracy does not necessarily mean that the model performs better on novel classes than on the seen classes as these performances are not directly comparable. Namely, the model can assign seen class data to novel classes and in that way reduce the recall of seen classes. Thus, the model can have higher recall on the novel classes mainly because the model finally learns to assign most of the ambiguous samples to novel classes. Thus, the best metric is overall accuracy on all classes.
> >
> > **RE: In parallel to the zero margin and fixed margin experiments have the authors conducted any experiments by decaying the margin for CE loss with a predefined schedule (linear, sigmoid, cosine, etc)? Even though this approach involves some hyperparameter tuning it would reduce the computation needed to compute the uncertainty adaptive margin.**
> >
> > We thank the reviewer for the question. We note that (1) our uncertainty adaptive margin does not need additional computational cost since it is estimated directly from the outputs of the softmax function, and (2) it is parameter-free compared to predefined schedules.
> > However, we additionally experimented with linear margin scheduling proposed by the reviewer. The result shown below demonstrate that  (1) linear scheduling is suboptimal compared to our uncertainty based margin and (2) linear scheduling performs better than ORCA-ZM and ORCA-FNM, proving that using a similar trend (large margin during the start of the training and small margin in the end) helps regardless of specific implementation. We included results in the Appendix in Table 12.
> >
> > | Margin Setup  | Seen | Novel | All  |
> > |-------------|------|-------|------|
> > | Linear     | 65.1 | 40.7  | 45.9 |
> > | Uncertainty      | 66.9 | 43.0  | 48.1 |

---

### Official Review · Reviewer_VV71 · 2021-11-02

**Correctness:** 3
**Technical Novelty And Significance:** 4
**Empirical Novelty And Significance:** 3
**Recommendation:** 6
**Confidence:** 5

**Main Review:**

# Strong points-
- By defining a new, realistic problem, the authors provide a direction for researchers to move forward with a persuasive and structured explanation.
- This paper contains a wide range of experiments to support the authors' claims. Although there is no clear competitor in the defined problem, authors adopted various methods from relevant fields, with slight appropriate changes. Also, they provide ablation study results to validate the proposed method.

# Weak points
- Compared to the detailed technical explanation of the proposed method, the solid explanation of the statements and intuition is insufficient.
- As the authors define a new problem, open-world SSL, the experimental setting is somewhat arbitrary.

# Questions
- While uncertainty adaptive margin takes a key role in ORCA, a detailed explanation of how it works doesn’t seem to suffice. Readers would not be able to understand with only intuitive explanations and references, so a more detailed explanation is necessary. For example, why eq.3 means the large margin of seen classes in early training epochs?
- In the experiment settings, authors set the ratio of seen/novel classes to 50%. Can it be said that this setting represents a real-world problem?
- Though there are no competitors to open-world SSL, baseline algorithms like FixMatch and DS3L show somewhat low performances than its original paper with conventional SSL settings. For the authors' method to be more convincing, it seems necessary to show that the ORCA does not fall behind the baseline in the existing SSL problem (without any novel class).


**Summary Of The Paper:**

In this paper, the authors define a problem with a more realistic perspective in semi-supervised learning, open-world semi-supervised learning, which considers a situation that unseen classes are included in unlabeled and test datasets. The authors propose a method called ORCA, which tries to solve the problem caused by open-world semi-supervised learning. The main function of the ORCA is balancing intra-class variation between seen and novel classes with 3 kinds of loss terms.


**Summary Of The Review:**

Although the paper has some weaknesses such as comparison in the conventional SSL setting, the paper proposes a new problem that considers a more realistic situation and the proposed method seems to work properly for the designed setting.

---

> ### Author Response · Authors · 2021-11-17
> **Response to Reviewer VV71**
>
> We thank the reviewer for the valuable comments and positive evaluation. We are glad that the reviewer acknowledges (1) the significance of our contributions, (2) a new realistic setting that we propose, and (3) our extensive experiments. We hope that our response helps in increasing the reviewer's confidence in our work.
>
> **RE: Compared to the detailed technical explanation of the proposed method, the solid explanation of the statements and intuition is insufficient.**
>
> We thank the reviewer for the comment. Besides our detailed technical explanation, we now provided a better intuitive explanation for parts of the text that the reviewer emphasized (please see below).
>
> **RE: As the authors define a new problem, open-world SSL, the experimental setting is somewhat arbitrary.**
>
> In the strong points of our work the reviewer gave a very insightful comment about our experimental setting, “Although there is no clear competitor in the defined problem, authors adopted various methods from relevant fields, with slight appropriate changes. Also, they provide ablation study results to validate the proposed method.” Indeed, we adopted state-of-the-art methods from all related settings (conventional SSL, safe SSL, novel class discovery, open-set recognition) that could be applied to solve open-world SSL.
>
> **RE: While uncertainty adaptive margin takes a key role in ORCA, a detailed explanation of how it works doesn’t seem to suffice. Readers would not be able to understand with only intuitive explanations and references, so a more detailed explanation is necessary. For example, why eq.3 means the large margin of seen classes in early training epochs?**
>
> We thank the reviewer for asking for clarification. The answer lies in equation (4). Namely, we use uncertainty as the margin. In early training epochs, uncertainty is large leading to a large margin, while as the training proceeds the uncertainty becomes smaller which leads also to smaller margin. In response to the reviewer’s feedback, we have clarified this better and reorganized the paragraph 3.3. We hope that the reviewer will find our explanation easier to follow now.
>
> **RE: In the experiment settings, authors set the ratio of seen/novel classes to 50%. Can it be said that this setting represents a real-world problem?**
>
> We have experiments with different ratios of seen/novel classes in the Appendix C (please see Figure 4). In that experiment, we already evaluated ORCA with different ratios (10%, 30%, 50%, 70%, 90%). The results showed that  even with the 90% of novel classes, ORCA achieves higher performance than all the baselines with 50% of novel classes. Thus, ORCA is applicable with different ratios of seen/novel classes.
>
> **RE: Though there are no competitors to open-world SSL, baseline algorithms like FixMatch and DS3L show somewhat low performances than its original paper with conventional SSL settings. For the authors' method to be more convincing, it seems necessary to show that the ORCA does not fall behind the baseline in the existing SSL problem (without any novel class).**
>
> Low performance of conventional semi-supervised methods with the presence of novel classes and out-of-distribution examples in the unlabeled data is a well known problem in the community (e.g., [1]). To address this important problem, we proposed a more realistic open-world SSL setting which is the focus of our work. To add more evidence, we repeated the experiment in Table 2. in standard supervised learning setting  with no unseen classes and ORCA achieves 60.3 while FixMatch achieves 63.1 accuracy. Note that FixMatch achieves lower accuracy than in the original paper because we use ResNet-18 backbone in all our experiments, compared to WRN-28-8  used in the paper (we changed the backbone to ensure same backbone across all methods). This shows that ORCA achieves only  slightly worse performance in the standard setting than FixMatch and can be used in standard SSL setting. However, in standard SSL setting using FixMatch is expectedly optimal and we do not claim or expect to outperform FixMatch. In the future work, it would be interesting to explore augmentation strategies used in FixMatch to further enhance ORCA (although these are not generally applicable, for example to single-cell data) .
>
> [1] Oliver et al. Realistic Evaluation of Deep Semi-Supervised Learning Algorithms. NeuriIPS ‘18

---

### Official Review · Reviewer_YZto · 2021-11-02

**Correctness:** 4
**Technical Novelty And Significance:** 2
**Empirical Novelty And Significance:** Not applicable
**Recommendation:** 6
**Confidence:** 3

**Main Review:**

Overall, this paper is well written. The paper is well motivated. The idea of adaptive margin and estimating intra-class variance using uncertainty is novel to me.

I have a few questions regarding the experiments:

- In Figure 3 (right), the accuracy of ORCA-ZM drops significant at the 140 epoch. The authors explained that this shows ORCA-ZM is not able to reduce intra-class variance. However, it's not clear why the decay of learning rate will trigger the performance drop.

- I suggest replacing "improvement" in Table 2 with "relative improvement" for the readers convenience.

- The proposed margin term is neat as described in Eq 4. It would be good if the authors could provide some analysis regarding L_S when the margin is Eq (4). For example, how will the margin mitigate the bias?

**Summary Of The Paper:**

This paper studies a new setting for open-world semi-supervised learning. This setting extends the typical semi-supervised learning by considering unseens classes in the test set. This paper introduces a uncertainty adaptive margin mechanism for this new problem. The results are evaluated on multiple datasets. The results are promising.

**Summary Of The Review:**

This paper is very well written and well motivated. I recommend accepting this paper for publication if the authors could address a few issues described above.

---

> ### Author Response · Authors · 2021-11-17
> **Response to Reviewer YZto**
>
> We thank the reviewer for the positive evaluation of our work and for recommending to accept the paper based on our feedback. We hope that our response helps to increase the reviewer's confidence and we are happy to provide additional feedback if needed.
>
> **RE: In Figure 3 (right), the accuracy of ORCA-ZM drops significant at the 140 epoch. The authors explained that this shows ORCA-ZM is not able to reduce intra-class variance. However, it's not clear why the decay of learning rate will trigger the performance drop.**
>
> We thank the reviewer for the insightful question. First, please note that our explanation refers to the novel classes where during the whole training process ORCA-ZM achieves low performance. Second, the reason why we see that decay of learning rate triggers performance drop on seen classes is that small learning rate tends to lead to overfit issues [1]. In ORCA-ZM the gap between intra-class variance of seen and novel classes is large and it leads to a very noisy estimation of pairwise pseudo-labels, and the noisy labels usually deteriorate only in the last stage with small learning rate [2]. We included an explanation in the revision and cited these works.
>
> [1] Li, Yuanzhi, Colin Wei, and Tengyu Ma. "Towards Explaining the Regularization Effect of Initial Large Learning Rate in Training Neural Networks." Advances in Neural Information Processing Systems 32 (2019).
>
> [2] Song, Jiaming, et al. "Robust and on-the-fly dataset denoising for image classification." European Conference on Computer Vision. Springer, Cham, 2020.
>
> **RE: I suggest replacing "improvement" in Table 2 with "relative improvement" for the readers convenience.**
>
> We thank the reviewer for this suggestion and we changed the paper accordingly.
>
> **RE: The proposed margin term is neat as described in Eq 4. It would be good if the authors could provide some analysis regarding L_S when the margin is Eq (4). For example, how will the margin mitigate the bias?**
>
> We thank the reviewer for appreciating the idea of our uncertainty adaptive margin approach. The margin mitigates the bias by enforcing similarly large intra-class variance of seen and novel classes. This is crucial to ensure high pseudo-labels quality and avoid generating error-prone pseudo-labels which assign novel classes to seen due to the difference in variance. To confirm this, we performed an analysis of the quality of pseudo-labels between L_s with margin (ORCA) and L_s without margin (ORCA-ZM) in Figure 5 in the Appendix. This analysis directly shows that L_s with margin (Eq 4) significantly improves the quality of generated pseudo-labels compared to L_s without margin.

---

### Official Review · Reviewer_zKw3 · 2021-11-02

**Correctness:** 3
**Technical Novelty And Significance:** 3
**Empirical Novelty And Significance:** 3
**Recommendation:** 6
**Confidence:** 5

**Main Review:**

Pros:

+ This paper is well-written and easy to follow. The motivation is clear and the overall organization is good.

+ A new setting is proposed in the community of semi-supervised learning, which jointly considers semi-supervised learning, open-set learning, and novel class discovery. This is a difficult but very practice problem in the real world.

+ A simple but effective approach, Uncertainty Based Adaptive Margin, is proposed to address the introduced setting. It can avoid the model quickly converge the seen classes so that the model can well recognize both seen and unseen classes.

+ Extensive experiments are provided to verify the effectiveness of the proposed method. In addition, this paper makes great contributions to implement existing SSL and novel class discovery methods into open-world learning.

Cons:

- This paper states that the proposed method can estimate the number of unseen classes. However, it seems that it should first estimate a rough number with DTC. How about the results of set a larger number of unseen classes (e.g., 200 for CIFAR100), learn the proposed model and then estimate the number of unseen classes?

- I appreciate that the authors implemented existing SSL and novel class discovery methods on the proposed setting. However, their implementation details are not very clear to me. Please introduce how to reproduce them in the supplementary.

- For the estimation of uncertainty, why not use other methods? Such as the entropy, and MC-Drop [A]? In addition, for the fixed negative margin, does 0.5 produce the best results? If not, the authors should compare the proposed method with different values of the negative margin.

- The proposed pairwise loss and regularization loss are not novel. We can find many semi-supervised learning and clustering methods using these two losses.

- In addition, for the pairwise loss, this paper selects the nearest neighbor as the positive candidate. However, in practice, this could introduce many false-negative pairs when the number of classes is large and the training batch size is small. I think this will be an issue for the proposed method especially when there are many classes. For example, the proposed method may meet problems when learning on the large ImageNet which has 1000 classes. Indeed, I think this is a long-standing problem that our community should consider and this may not be addressed in this paper.

- For self-supervised learning, I would like to see the results of using different self-supervised learning methods, such as SimCLR, MOCOV2, and RotationNet. It is very interesting that [B] found that RotationNet achieves better results than SIMCLR and MOCOV2 for novel class discovery. Therefore, I am curious about that if this phenomenon happens in the proposed setting.

- I think the main contribution is the proposed adaptive margin method. Therefore, this approach can be applied to other novel class discovery methods. How about applying the proposed margin method to RankingStatic?

- I think one important result is missed in the paper. We assume that the samples of novel classes and samples of seen classes are separated in the unlabeled data. That is, for the unlabeled data, we know which samples are from novel classes or seen classes but we do not know the class labels. I think this is could one upper bond for the proposed setting. In addition, under such an assumption, can the proposed method improve the results? Or, we only need simple pairwise loss and regularization loss (also RankingStatics)?

- Another two important experiments are missing. (1) Applying existing open-set methods for the introduced setting. (2) Applying the proposed method on the open-set recognition setting.

- At last, I strongly recommend the authors release the source code during submission. (1) This is the first work that studies the open-world setting. (2) I know that this paper was submitted to several venues and many researchers know this paper/setting. I also found several works that follow this setting and compare this work. Thus, providing the source code could be a good start for this new task.

[A] Y. Gal and Z. Ghahramani. Dropout as a Bayesian approximation: Representing model uncertainty in deep learning. In ICML, 2016

**Summary Of The Paper:**

This paper considers the problem of open-world learning, where the labeled data are from seen classes and unlabeled data are from both seen and novel classes. To address this problem, this paper presents an Uncertainty Based Adaptive Margin method for learning the joint classifier of seen and unseen classes. The proposed method can avoid the model assigning samples of novel classes to the seen classes. Experiments on several datasets show the effectiveness of the proposed method.

**Summary Of The Review:**

Overall, I like this work especially the proposed setting. Although the novelty is not very strong, this is the first step for open-world learning. Also, there are some concerns and missing experiments that should be addressed during the rebuttal. I think this is a promising and interesting task so that the first work should design a well-considered setting and provide extensive comparisons for the following works.

---

> ### Author Response · Authors · 2021-11-17
> **Response to Reviewer zKw3 (part 1)**
>
> We thank the reviewer for the valuable feedback and positive evaluation of our work. We are glad that the reviewer likes our paper and finds the proposed setting important, difficult and practical. We are also glad that the reviewer appreciates the simplicity and effectiveness of our method and extensiveness of our experiments. As we explain in more detail below, many additional experiments the reviewer is asking are already provided in our paper/Appendix. In addition to these, we conducted further validation which added more evidence to our method. We hope that these results and our response will help in increasing the reviewer's confidence.
>
> **RE: This paper states that the proposed method can estimate the number of unseen classes. However, it seems that it should first estimate a rough number with DTC. How about the results of set a larger number of unseen classes (e.g., 200 for CIFAR100), learn the proposed model and then estimate the number of unseen classes?**
>
> We ran ORCA with 200 classes on CIFAR100 and we report the performance in the table below. ORCA retains high performance even when initialized with 200 classes. In total, there were 139 activated heads, among which 102 heads contain more than 50 samples. The high performance also indicates that most of the additional clusters contain smaller subclasses of the correct class and belong to meaningful clusters.  We additionally ran ORCA with 400 classes and show that ORCA retains high accuracy. We included the results in Table 13 in Appendix.
>
> | Num of Heads  | Seen | Novel | All  |
> |-------------|------|-------|------|
> | 100      | 66.9 | 43.0  | 48.1 |
> | 200     | 66.5 | 41.7  | 47.2 |
> | 400     | 65.3 | 39.6  | 43.4 |
>
> **RE: I appreciate that the authors implemented existing SSL and novel class discovery methods on the proposed setting. However, their implementation details are not very clear to me. Please introduce how to reproduce them in the supplementary.**
>
> We thank the reviewer for asking for the clarification. In response to reviewer’s feedback, we added a paragraph with implementation details for baselines in the Appendix A.
>
> **RE: For the estimation of uncertainty, why not use other methods? Such as the entropy, and MC-Drop [A]? In addition, for the fixed negative margin, does 0.5 produce the best results? If not, the authors should compare the proposed method with different values of the negative margin.**
>
> We believe that the reviewer missed our experiment with different margin values (please see Table 6 in the  Appendix). Margin 0.5 indeed produces the best results. MC-Drop is dependent on using Dropout in the network architecture, which we do not use so it is not applicable. Compared to MC-Drop, our approach is architecture-agnostic and does not require any additional computation. Entropy is a nonlinear projection of our current approach and achieves suboptimal performance compared to our uncertainty based margin. We report results on the CIFAR-100 dataset below and included it in the Appendix in Table 12.
>
> | Margin Setup  | Seen | Novel | All  |
> |-------------|------|-------|------|
> | Entropy     | 66.5 | 41.7  | 46.5 |
> | Uncertainty      | 66.9 | 43.0  | 48.1 |
>
> **RE: The proposed pairwise loss and regularization loss are not novel. We can find many semi-supervised learning and clustering methods using these two losses.**
>
> We thank the reviewer for the comment. We agree that the pairwise loss and regularization loss are not novel parts of our objective function and we do not claim the novelty in that part. We cite related work for both losses. However, as we show these losses are not sufficient to solve challenging the open-world learning setting proposed in our work. As demonstrated in our experiments, naive combination of pairwise loss, regularization loss and cross-entropy loss (referred to as ORCA-ZM) does not achieve satisfactory performance.  The main technical contribution of our approach lies in introducing uncertainty based adaptive margin which is proposed for the first time in our work and it is shown to be a key component to solving challenging open-world learning setting.

---

> > ### Author Response · Authors · 2021-11-17
> > **Response to Reviewer zKw3 (part 2)**
> >
> > **RE: In addition, for the pairwise loss, this paper selects the nearest neighbor as the positive candidate. However, in practice, this could introduce many false-negative pairs when the number of classes is large and the training batch size is small. I think this will be an issue for the proposed method especially when there are many classes. For example, the proposed method may meet problems when learning on the large ImageNet which has 1000 classes. Indeed, I think this is a long-standing problem that our community should consider and this may not be addressed in this paper.**
> >
> > We thank the reviewer for the question about handling a larger number of classes. There are a number of strategies to deal with the larger number of classes and the simplest solution is to increase the batch size. Another possibility is to accumulate several batches when computing positive candidates. We agree with the reviewer that achieving high performance with a large number of classes (e.g. ImageNet with 1000 classes) is a long-standing problem of the community.
> >
> > **RE: For self-supervised learning, I would like to see the results of using different self-supervised learning methods, such as SimCLR, MOCOV2, and RotationNet. It is very interesting that [B] found that RotationNet achieves better results than SIMCLR and MOCOV2 for novel class discovery. Therefore, I am curious about that if this phenomenon happens in the proposed setting.**
> >
> > We thank the reviewer for the suggestion. In response, we experimented with switching the self-supervised algorithm to RotationNet. We find that additional improvements can be achieved using RotationNet. Results on the CIFAR-100 dataset are shown in Table below. In response to the reviewers feedback, we included the result in the Appendix as Table 11.
> >
> > | SSL         | Method   | Seen | Novel | All  |
> > |-------------|----------|------|-------|------|
> > | SimCLR      | ORCA-ZM  | 55.2 | 32.0  | 34.8 |
> > | SimCLR     | ORCA-FNM | 58.2 | 40.0  | 44.3 |
> > | SimCLR      | ORCA     | 66.9 | 43.0  | 48.1 |
> > | RotationNet | ORCA-ZM  | 57.6 | 33.4  | 37.9 |
> > | RotationNet | ORCA-FNM | 59.2 | 42.7  | 47.3 |
> > | RotationNet | ORCA     | 70.0 | 45.6  | 51.4 |
> >
> > **RE: I think the main contribution is the proposed adaptive margin method. Therefore, this approach can be applied to other novel class discovery methods. How about applying the proposed margin method to RankingStatic?**
> >
> > The gap between variance of seen and novel classes which adaptive margin solves is not an issue in novel class discovery methods (such as RankStats) so these methods are not expected to benefit from applying adaptive margin. The reason is that the novel class discovery problem is an easier problem than open-world SSL and it is a priori known which classes are seen or novel.
> >
> > **RE: We assume that the samples of novel classes and samples of seen classes are separated in the unlabeled data. That is, for the unlabeled data, we know which samples are from novel classes or seen classes but we do not know the class labels. I think this is could one upper bond for the proposed setting. In addition, under such an assumption, can the proposed method improve the results? Or, we only need simple pairwise loss and regularization loss (also RankingStatics)?**
> >
> > Simple pairwise loss and regularization loss as done in RankStats are indeed sufficient to solve novel class discovery which is an easier problem compared to open-world SSL. Thus, the adaptive margin is not expected to bring benefits in that setting since it solves the gap between variance of seen and novel classes which is not a problem in novel class discovery.

---

> > > ### Author Response · Authors · 2021-11-17
> > > **Response to Reviewer zKw3 (part 3)**
> > >
> > > **RE: Another two important experiments are missing. (1) Applying existing open-set methods for the introduced setting. (2) Applying the proposed method on the open-set recognition setting.**
> > >
> > > We thank the reviewer for the comment, but we believe that the reviewer missed our comparison to state-of-the-art open-set recognition baseline CGDL [1] (Table 2). ORCA achieves substantial performance gains compared to CGDL, for example, 105% improvement on CIFAR-100. The reason is that open-set recognition methods fail to recognize unseen classes as out-of-distribution samples when the domain gap of seen and unseen classes is not significant. Regarding the reverse direction, we note that the focus of this paper is on the new open-world SSL setting, while open-set recognition is a different problem (evaluated by adding samples with a domain gap as OOD samples) and it does not fit in the scope of this open-world semi-supervised learning work.
> > >
> > > [1]  Sun et al. Conditional Gaussian Distribution Learning for Open Set Recognition. CVPR ‘20
> > >
> > > **RE: At last, I strongly recommend the authors release the source code during submission. (1) This is the first work that studies the open-world setting. (2) I know that this paper was submitted to several venues and many researchers know this paper/setting. I also found several works that follow this setting and compare this work. Thus, providing the source code could be a good start for this new task.**
> > >
> > > We thank the reviewer for showing interest in the source code of our method. As stated in the submission, we will definitely release our code so that others can reproduce and build upon our method.

---

### Official Review · Reviewer_fpiF · 2021-11-02

**Correctness:** 3
**Technical Novelty And Significance:** 2
**Empirical Novelty And Significance:** 3
**Recommendation:** 6
**Confidence:** 4

**Main Review:**

Strength:

This paper proposed a new setting for SSL. Simultaneously training classifier for seen classes and discovering novel classes from unlabeled data is novel.

Comparison with existing methods adapted to the new setting is competitive.

Weakness:

The expected number of novel classes is a strong assumption for open-world SSL. Inferring the number of classes is only briefly mentioned in the paper. I would like to see more details.

The uncertainty in Eq(4) is not explained well. I am wondering why other choices, e.g. entropy, is not selected. Moreover, is the max_k is among the known classes or over all classes? If it is over all classes I do not see how the classifier W_k, k\in Cu is trained.

I do not see any term that can avoid assigning unlabeled data to seen classes. For example Eq(5) can still be minimized if all unlabeled pairs are assigned to the same seen classes.

Regularizing the average posterior w.r.t. a prior distribution is somehow a strong assumption in that the distribution for unseen classes are known in advance.

Pre-training could substantially improve the quality of feature representation. As a result, the pair-wise objective might heavily rely on the pre-training. I think it is worth discussing or providing experiments to validate this point.


**Summary Of The Paper:**

This work proposed a novel setting for SSL where unlabeled data contains novel categories and it aims to simultaneously train classifiers for seen classes and discover novel classes. This is achieved by introducing pairwise loss and regularizing with prior class distribution. Experiments are carried out on synthesized data splits and demonstrated good results.


**Summary Of The Review:**

This paper provides a new perspective into SSL. But the designs are not fully explained and some remains unjustified, e.g. the prior distribution for all classes and the number of novel classes.

---

> ### Author Response · Authors · 2021-11-17
> **Response to Reviewer fpiF (part 1)**
>
> We thank the reviewer for the valuable comments and for acknowledging the novelty of our work and our competitive experimental results. We address the reviewer's concerns below and we hope that our response helps to increase the reviewer's confidence.
>
> **RE: The expected number of novel classes is a strong assumption for open-world SSL. Inferring the number of classes is only briefly mentioned in the paper. I would like to see more details.**
>
> We agree with the reviewer and we do not assume that the number of novel classes is known. ORCA can be initialized with a large number of classes, and automatically infer number of classes by never activating additional classification heads. In addition to our experiments in the paper with an estimated number of novel classes  (124 classes), we report the results with a large number of classes for CIFAR-100 in the table below. ORCA achieves high performance even with 200 and 400 classification heads and outperforms all baselines even when they use the correct number of classes (100).  Please notice that this is a remarkable feat, since clustering and novel class discovery methods assume that the number of clusters is known ahead of time. We included the results in Table 13 in Appendix.
>
> | Num of Heads  | Seen | Novel | All  |
> |-------------|------|-------|------|
> | 100      | 66.9 | 43.0  | 48.1 |
> | 200     | 66.5 | 41.7  | 47.2 |
> | 400     | 65.3 | 39.6  | 43.4 |
>
> **RE: The uncertainty in Eq(4) is not explained well. I am wondering why other choices, e.g. entropy, is not selected. Moreover, is the max_k is among the known classes or over all classes? If it is over all classes I do not see how the classifier W_k, k\in Cu is trained.**
>
> We thank the reviewer for the question. Entropy is just a nonlinear projection of our current approach and achieves suboptimal performance compared to our uncertainty based margin. We report results with entropy on the CIFAR-100 dataset in the table below and we included it in the Appendix as Table 12.
>
> | Margin Setup  | Seen | Novel | All  |
> |-------------|------|-------|------|
> | Entropy     | 66.5 | 41.7  | 46.5 |
> | Uncertainty      | 66.9 | 43.0  | 48.1 |
>
> Max_k is among all classes. We made it clear now in the revision. Note that here we only need an estimation about how confident the model is on the unlabeled data to adjust the margin accordingly. $W_k, k\in C_u$ is trained through the pairwise objective in which the number of classification heads corresponds to the total number of classes (seen+estimated/expected novel).
>
> **RE: I do not see any term that can avoid assigning unlabeled data to seen classes. For example Eq(5) can still be minimized if all unlabeled pairs are assigned to the same seen classes.**
>
> Regularization term avoids a trivial solution of assigning all examples in the same class (please see Section 3.5). This term corresponds to the maximum entropy regularization and it has been used in pseudo-labeling based SSL [1], deep clustering methods [2] and training on noisy labels [3] to prevent the class distribution from being too flat.
>
> [1] Arazo et al. Pseudo-labeling and confirmation bias in deep semi-supervised learning. IJCNN ‘20
>
> [2] Van Gansbeke et al. SCAN: Learning to classify images without labels. CVPR ‘20
>
> [3] Tanaka et al. Joint optimization framework for learning with noisy labels.CVPR ‘18
>
> **RE: Regularizing the average posterior w.r.t. a prior distribution is somehow a strong assumption in that the distribution for unseen classes are known in advance.**
>
> We agree with the reviewer that knowing prior distribution is a strong assumption and we do not assume that the prior distribution is known in advance. In all our experiments we regularize the term with maximum entropy and show that this regularization works well for balanced and unbalanced data (please see Table 4 and Table 9 in the Appendix for additional experiments with unbalanced data). However, in rare cases when prior probability distribution is known, one can regularize the objective towards known prior distribution. In response to the reviewer’s feedback, we modified the text in Section 3.5 to make this more clear.

---

> > ### Author Response · Authors · 2021-11-17
> > **Response to Reviewer fpiF (part 2)**
> >
> > **RE: Pre-training could substantially improve the quality of feature representation. As a result, the pair-wise objective might heavily rely on the pre-training. I think it is worth discussing or providing experiments to validate this point.**
> >
> > We thank the reviewer for the important point. We want to emphasize that three analyses in our work directly show that ORCA’s performance is not due to pretraining:
> >
> > 1. We pretrained ORCA and all baselines in our work using SimCLR. Thus, every method in Table 2 is pretrained with SimCLR to ensure that benefits of ORCA are coming solely from our objective function and not from the pretraining. ORCA outperforms baselines by 108% on CIFAR-100 dataset and 93% on ImageNet.
> >
> > 2. We incorporated SimCLR baseline in Table 2 which directly evaluates  the performance of the pretraining step. In particular, we run SimCLR and then run clustering on pretrained features. ORCA achieves improvement over this baseline by 116% on CIFAR-100 and 111% on ImageNet.
> >
> > 3. Finally, on single-cell dataset we do not use any pretraining strategy and ORCA starts from randomly initialized weights.
> >
> > We believe that all these analyses strongly support our claim that ORCA’s improvements are directly caused by our objective function and not by pretraining

---

> > > ### Comment · Reviewer_fpiF · 2021-11-29
> > > **Details of pretraining**
> > >
> > > As pretraining have a huge impact on the performance. I recommend the authors to provide more details of SimCLR pretraining, e.g. on what dataset, hyperparameters, etc. This will allow fair comparisons for follow-up works.

---

> > > > ### Author Response · Authors · 2021-11-29
> > > > **Response to Reviewer fpiF**
> > > >
> > > > We thank the reviewer for the suggestion. We use the OpenSelfSup toolbox (https://github.com/open-mmlab/OpenSelfSup) and we set all hyperparameters to default for both CIFAR and ImageNet datasets. We pretrain the backbone on the whole dataset for each experiment, and we additionally include the experiment in which we pretrain only on the labeled subset of the data in the Appendix (Table 10). We use the same pretrained backbone for ORCA and all other baselines. We will include these details and we will make the pretrained models publicly available to make sure that follow-up works can easily compare to our work.

---

### Decision · Program_Chairs · 2022-01-20

**Decision:**

Accept (Poster)

**Comment:**

This paper is proposed to address a novel but practical setting that the test set consists of both seen and unseen classes of the training set. To tackle the crucial challenge of distribution mismatch between the inlier and outlier features, the authors proposed a new method named ORCA by grouping similar instances to enlarge the class-wise margin for de-biasing. The experimental results on ImageNet have shown the proposed ORCA has significantly outperformed baselines in both inlier classification and outlier detection. The whole paper is written with clear logic and is easy to follow. Moreover, such a new setting may bring more inspiration to the community.

---

> ### Public Comment · ~Michelle_Robinson1 · 2022-10-09
> **:)**
>
> Thank you :) I discovered a really fascinating website last week at [writinguniverse.com/compare-and-contrast-essay-topics/](https://writinguniverse.com/compare-and-contrast-essay-topics/) This enables me to simply do my college projects on time. If you'd want to do the same, visit this website and finish your tasks quickly. and inspire your parents.